# On the Frequency Bias of Generative Models

**Katja Schwarz**          **Yiyi Liao**          **Andreas Geiger**

Autonomous Vision Group
University of Tübingen and MPI for Intelligent Systems
`{firstname.lastname}@uni-tuebingen.de`

## Abstract

The key objective of Generative Adversarial Networks (GANs) is to generate new data with the same statistics as the provided training data. However, multiple recent works show that state-of-the-art architectures yet struggle to achieve this goal. In particular, they report an elevated amount of high frequencies in the spectral statistics which makes it straightforward to distinguish real and generated images. Explanations for this phenomenon are controversial: While most works attribute the artifacts to the generator, other works point to the discriminator. We take a sober look at those explanations and provide insights on what makes proposed measures against high-frequency artifacts effective. To achieve this, we first independently assess the architectures of both the generator and discriminator and investigate if they exhibit a frequency bias that makes learning the distribution of high-frequency content particularly problematic. Based on these experiments, we make the following four observations: 1) Different upsampling operations bias the generator towards different spectral properties. 2) Checkerboard artifacts introduced by upsampling cannot explain the spectral discrepancies alone as the generator is able to compensate for these artifacts. 3) The discriminator does not struggle with detecting high frequencies per se but rather struggles with frequencies of low magnitude. 4) The downsampling operations in the discriminator can impair the quality of the training signal it provides. In light of these findings, we analyze proposed measures against high-frequency artifacts in state-of-the-art GAN training but find that none of the existing approaches can fully resolve spectral artifacts yet. Our results suggest that there is great potential in improving the discriminator and that this could be key to match the distribution of the training data more closely.

## 1 Introduction

In recent years, unconditional Generative Adversarial Networks (GANs) have achieved impressive photo-realism for image synthesis tasks. While this has hampered the identification of generated images based on visual cues, multiple recent works show that it is straightforward to distinguish real and generated images based on their high-frequency content [7–9, 14, 22, 34, 38]. This has aroused considerable interest because it reveals a fundamental problem in state-of-the-art GANs: Existing approaches evidently struggle to learn the correct data distribution. While GAN training is notoriously hard, learning the distribution of high-frequency content is particularly problematic [7–9, 14, 22, 34, 38]. This indicates a systematic problem in existing approaches that could make training suboptimal. For example, if generating high-frequencies is difficult for the generator but detecting them is straightforward for the discriminator this imbalance could impair the stability of the training. Conversely, if the discriminator struggles to detect high frequencies generating fine details also becomes more difficult which could impede convergence. Therefore, we argue that it is important to better understand the expressivity of both the generator and the discriminator. Indeed, numerous works suggest that the architecture of the generator can hamper the generation of high

35th Conference on Neural Information Processing Systems (NeurIPS 2021).

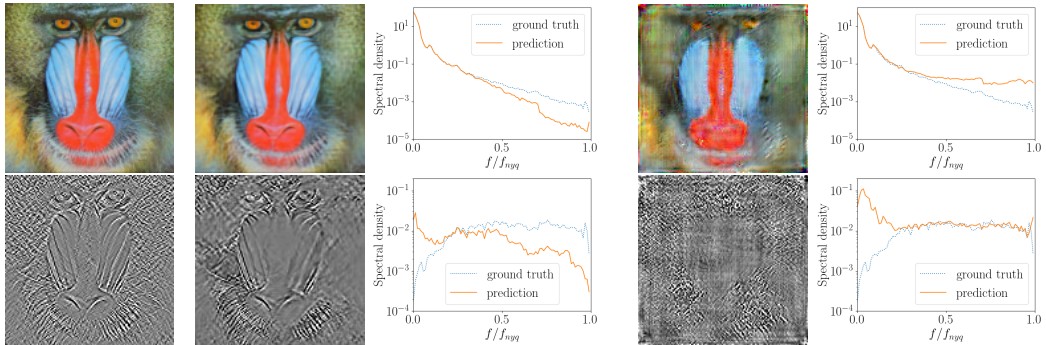

(a) Ground truth | (b) Reconstruction of a generator with bilinear upsampling using pixel-level supervision | (c) Reconstruction guided by a discriminator with BlurPool downsampling

Figure 1: **Spectral Properties of Generator and Discriminator.** (a) We show a natural image and a high-pass filtered version (amplified for clarity) to illustrate the networks' behavior for both low and high magnitudes at high frequencies. (b) Our experiments reveal that bilinear upsampling biases the generator towards generating data with little high-frequency content, regardless of the magnitude. (c) The discriminator shows no such bias but provides better guidance for frequencies with high magnitude. The results further reveal artifacts due to downsampling that impair the training signal. We include the training details for both experiments in the supplementary.

frequencies [4, 7, 9, 10, 14, 20]. However, another line of works question the quality of the training signal provided by the discriminator [5, 10, 15]. But is there indeed a *frequency bias* that prevents learning high frequencies in existing GAN models?

This is a non-trivial question as GAN training involves two players where architectures for both the generator and discriminator, loss functions, as well as the dataset statistics can all affect the generated images. To narrow down potential factors, we first develop isolated testbeds for both the generator and discriminator. Then, we extend our findings to state-of-the-art GANs on large-scale datasets.

**Generator:** The majority of works attribute the high-frequency artifacts to the upsampling operations in the generator [4, 7, 9, 25, 38]. Upsampling typically follows a pre-defined scheme, e.g. bilinear or nearest neighbor interpolation, or zero insertion between pixels. While the latter introduces too many high frequencies, interpolation has been shown to reduce artifacts [4, 7]. But *is interpolation always a good scheme for upsampling?* Our experiments indicate that both bilinear and nearest neighbor upsampling bias the generator towards predicting little high-frequency content, see Fig. 1b. While zero insertion can be more flexible, it is prone to introducing checkerboard artifacts, i.e. too many high frequencies [4, 7, 25]. But *why do the learnable filters in subsequent layers not learn to compensate for these artifacts?* Depending on the training objective, checkerboard artifacts might be penalized only slightly. Our experiments evidence that when the loss function is sensitive to these artifacts the generator is indeed capable of compensating for them.

**Discriminator:** This leads us to question the quality of the training signal: *Can the discriminator detect high frequencies and provide the necessary supervision?* Chen et al. [5] argue that the discriminator cannot detect high-frequency information due to the downsampling operations. Our experiments corroborate that downsampling can introduce artifacts in the training signal. However, we also observe that the discriminator can indeed provide a meaningful training signal for the spectral statistics at high frequencies when their magnitude is large enough, e.g., Fig. 1c. Other works propose to train both the generator and discriminator in wavelet space [10] or add a discriminator on the spectrum of the images to reduce high-frequency artifacts [15]. Motivated by these approaches we ask: *Is it enough to consider only the spatial domain?* In agreement with these works, our results suggest that the discriminator can benefit from (additional) input in frequency-based domains. Further, our testbed yields insights on which of these measures is most effective and reveals that the training signal from the discriminator might remain problematic.

**Contributions:** We take a sober look at explanations for high-frequency artifacts in generated images and unify the efforts that have been done so far. In particular, we develop isolated testbeds for both the generator and the discriminator which allows us to analyze what makes existing efforts effective. The conclusions drawn by this paper shed new light on limitations of common design choices for both generator and discriminator: i) Bilinear and nearest neighbor upsampling bias the generator towards

predicting little high-frequency content. ii) Zero insertion is prone to producing checkerboard artifacts in the generated images. However, with a suitable loss function, the learnable filters of the generator can compensate for the artifacts. This indicates that the upsampling in the generator alone cannot explain the spectral discrepancies. iii) In general, the discriminator is able to detect high frequencies and provide supervision to learn the correct spectral statistics. However, while the exponential decay of the spectrum for natural images creates the impression of the discriminator being insensitive to high frequencies, it actually struggles with low magnitudes. iv) We find that all commonly used downsampling operations in the discriminator can impair the quality of the training signal.

Lastly, we demonstrate that these findings extend to full GAN training. In agreement with [15], we find that spectral discriminators can bring us one step closer to matching the spectral statistics. Nonetheless, generated images remain straightforward to classify based on their spectral statistics only. While recent works on GANs largely focus on improving the generator, e.g., [16, 18, 19], our findings suggest that the design of the discriminator plays an equally important role and deserves more attention in future work. We believe that our testbeds for both the generator and discriminator can be a useful tool for future investigations. We release our code and dataset at https://github.com/autonomousvision/frequency_bias.

## 2 Preliminaries

**GAN Training:** A generative adversarial network consists of a generator and a discriminator that are trained jointly in a 2-player game. Given latent variables $\mathbf{z} \sim p_z$, the generator synthesizes images while the discriminator tries to distinguish the synthesized images from real images $\mathbf{I} \sim p_{\mathcal{D}}$, sampled from data distribution $p_{\mathcal{D}}$. Let $G_\Theta$ and $D_\Phi$ denote a generator $G$ and a discriminator $D$ with parameters $\Theta$ and $\Phi$, respectively. The parameters of the models are updated in alternating steps using a non-saturating GAN objective [11] which is often combined with R1-regularization to stabilize training [24]

$$V(\Theta, \Phi) = \mathbb{E}_{\mathbf{z} \sim p_z} \left[ f(D_\Phi(G_\Theta(\mathbf{z}))) \right] + \mathbb{E}_{\mathbf{I} \sim p_{\mathcal{D}}} \left[ f(-D_\Phi(\mathbf{I})) - \lambda \|\nabla D_\Phi(\mathbf{I})\|^2 \right] \tag{1}$$

where $f(t) = -\log(1 + \exp(-t))$ and $\lambda$ controls the strength of the regularizer.

**Image Processing in the Frequency Domain:** The discrete 2D Fourier transform maps a gray scale image $\mathbf{I} \in \mathbb{R}^{H \times W}$ to the frequency domain:

$$\hat{\mathbf{I}}[k, l] = \frac{1}{HW} \sum_{x=0}^{H-1} \sum_{y=0}^{W-1} \exp^{-2\pi i \frac{x \cdot k}{H}} \exp^{-2\pi i \frac{y \cdot l}{W}} \cdot \mathbf{I}[x, y] \tag{2}$$

for $k = 0, \ldots, H-1$ and $l = 0, \ldots, W-1$. The power spectral density is estimated by the squared magnitudes of the Fourier components $\mathbf{S}[k, l] = |\hat{\mathbf{I}}[k, l]|^2$. Similar to [7, 8] we consider the reduced spectrum $\tilde{S}$, i.e. the azimuthal average over the spectrum in normalized polar coordinates $r \in [0, 1]$, $\theta \in [0, 2\pi)$

$$\tilde{S}(r) = \frac{1}{2\pi} \int_0^{2\pi} S(r, \theta) d\theta \quad \text{with} \quad r = \sqrt{\frac{k^2 + l^2}{\frac{1}{4}(H^2 + W^2)}} \quad \text{and} \quad \theta = \text{atan2}(k, l) \tag{3}$$

Since images are discretized in space, the maximum frequency is determined by the Nyquist frequency. For a square image, $H = W$, it is given by $f_{nyq} = \sqrt{k^2 + l^2} = H/\sqrt{2}$, i.e. for $r = 1$.

**Spectral Classifier:** To detect generated images, Dzanic et al. [8] propose to classify real and generated images based on their reduced spectrum. More specifically, they fit a power-law function to the tail of each reduced spectrum for frequencies above a given threshold $r_c = 0.75$ and train a binary classifier on the fit parameters of each spectrum.

## 3 Are high frequencies more difficult to generate?

In this section, we investigate if there is a frequency bias in the generator that impedes the generation of high frequencies. Therefore, we first consider the generator in an isolated setting assuming pixel-level supervision – otherwise, even with a perfect discriminator it would be impossible to generate a correct image. We extend our findings to the full GAN setting in Section 5.

There are two lines of argumentation in existing works: The first investigates artifacts that arise from the upsampling operations in the generator [4, 7, 9]. These works analyze the spectral properties of the generator predictions at convergence. We extend this analysis from the perspective of a frequency bias to see if low frequencies are learned earlier during training. The second line of works argues for a frequency bias of the learnable filters [14, 20]. Motivated by their analysis we investigate if learnable filters are at all able to compensate for the artifacts introduced by upsampling.

**Experimental Setting:** To isolate the generator from the discriminator we consider a conditional reconstruction task. In particular, we take 10 images from a dataset and pair them with 10 latent codes drawn from a normal distribution. Given a latent code, the generator is optimized to reconstruct the corresponding image with a pixel-wise L2-loss

$$L_I = \frac{1}{HW} \sum_{x=0}^{H-1} \sum_{y=0}^{W-1} \|G_\Theta(\mathbf{z})[x,y] - \mathbf{I}[x,y]\|_2^2 \tag{4}$$

We choose the generator from PGAN [16] because of its simple architecture comprising only convolutions and upsampling operations. Further, when combined with R1-regularization it trains stably in a GAN setting [24], allowing us to use the same generator in Section 5. We reduce the number of channels for faster training because our primary focus is on the spectral properties. As upsampling operations, we investigate bilinear and nearest neighbor interpolation, zero insertion, and reshaping in the channel dimension [29]. We ensure that all networks have a similar number of parameters. More details are provided in the supplementary.

**Datasets:** In natural images, the spectral density follows an exponential decay. To isolate the effects of frequency range and magnitude, we create a Toyset of images with two Gaussian peaks $\mathcal{N}(\mu_1, \sigma_1)$, $\mathcal{N}(\mu_2, \sigma_2)$ of equal magnitude in the spectrum, see Fig. 2a, 2b. Based on the Nyquist frequency $f_{nyq} = H/\sqrt{2}$, we create samples using $\sigma_1 = \sigma_2 = 1/\sqrt{2} f_{nyq}$ and draw $\mu_1 \sim \mathcal{U}(0.05 f_{nyq}, 0.15 f_{nyq})$ and $\mu_2 \sim \mathcal{U}(0.75 f_{nyq}, 0.85 f_{nyq})$. Together with a uniformly distributed phase, we apply the inverse Fourier transform to create images. We further test our setting on natural images with a downsampled version of CelebA [21]. Both datasets have a resolution of $64^2$ pixels.

**Evaluation Metrics:** On the spatial domain, we report PSNR on the RGB values and on the frequency domain we evaluate the reduced spectrum, see Eq. (3). To analyze frequency-dependent convergence we further visualize the evolution of the spectrum similar to [26]. Here, the x-axis denotes the (frequency) radius $r$ in normalized polar coordinates and the y-axis denotes the training iterations. The color corresponds to the relative error of the average predicted reduced spectrum wrt. the ground truth, where positive and negative values indicate too many and too few predicted frequencies, respectively. We clip the colorbar at 1, i.e., when the relative error exceeds $100\%$.

**Do generators exhibit a frequency bias?** The spectral evolution in Fig. 2 shows different behavior for bilinear and nearest neighbor upsampling compared to zero insertion and reshaping. Particularly on the Toyset, a generator with bilinear or nearest neighbor upsampling learns the lower frequencies earlier in training than the high frequencies, see Fig. 2c and 2d. While the generator with bilinear upsampling struggles with high frequencies throughout training, for nearest neighbor upsampling it eventually fits the high-frequency peak, suggesting that its bias towards little high-frequency content is not as strong. In contrast, a generator with zero insertion or reshaping learns both peaks approximately at equal speed but is prone to generating checkerboard artifacts as indicated by the large error at the highest frequency in Fig. 2e and 2f. Between the peaks of the toyset, where frequencies have small magnitudes, all methods struggle to match the statistics of the training data because the L2-loss penalizes errors at frequencies with low magnitude only slightly. This explains why for CelebA there is an overall trend towards learning lower frequencies earlier during training. The PSNR values in Table 1 support these findings in the spatial domain. The lack of high frequencies for bilinear and nearest neighbor upsampling manifests in a low PSNR, particularly for the Toyset which contains many high frequencies by construction.

**Can the generator learn to compensate for the artifacts?** Since the L2-loss is less sensitive to frequencies with low magnitudes (see supplementary for a formal derivation), we now add an L2-loss on the logarithm of the reduced spectrum:

$$L_S = \frac{1}{H/\sqrt{2}} \sum_{k=0}^{H/\sqrt{2}-1} \left\| \log\left(\tilde{S}\left(G_\Theta(\mathbf{z})\right)\right)[k] - \log\left(\tilde{S}(\mathbf{I})\right)[k] \right\|_2^2 \tag{5}$$

The logarithm penalizes errors at low magnitudes more strongly and can therefore reduce the checkerboard artifacts introduced by zero insertion and reshaping, see Fig. 3. Hence, given a suitable

| | Bilinear | | NN | | Zeros | | Reshape | |
| | | + $L_S$ | | + $L_S$ | | + $L_S$ | | + $L_S$ |
| --- | --- | --- | --- | --- | --- | --- | --- | --- |
| Toyset | 21.1 | 20.9 | 30.7 | 30.7 | 38.6 | 38.0 | 38.6 | 36.5 |
| CelebA | 34.9 | 37.0 | 40.9 | 39.8 | 42.4 | 43.6 | 42.3 | 40.9 |

Table 1: **PSNR** for different upsampling operations in the generator at resolution $64^2$ pixels.

| | AvgPool | | BlurPool | | Stride | | MLP | |
| | | + SD | | + SD | | + SD | | +SD |
| --- | --- | --- | --- | --- | --- | --- | --- | --- |
| Toyset | 16.3 | 19.2 | 23.3 | 15.6 | 23.3 | 24.1 | 26.2 | 46.5 |
| CelebA | 25.4 | 27.0 | 24.2 | 24.4 | 25.5 | 25.6 | 28.1 | 33.7 |

Table 2: **PSNR** for different downsampling operations in the discriminator at resolution $64^2$ pixels.

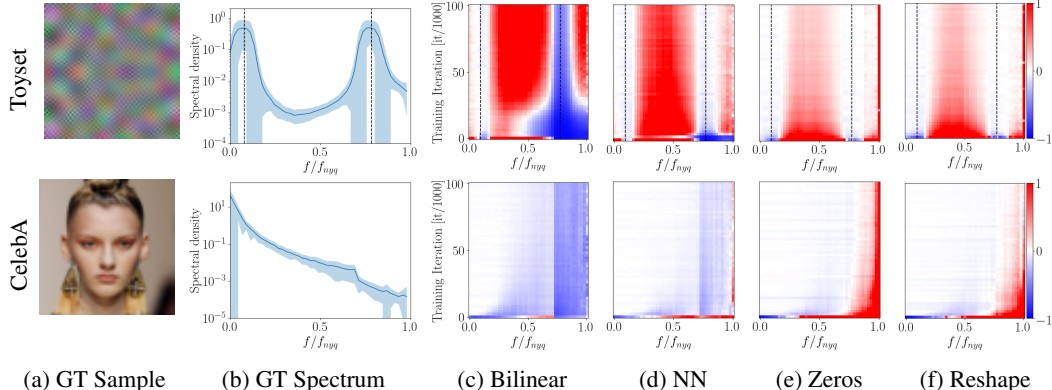

(a) GT Sample    (b) GT Spectrum    (c) Bilinear    (d) NN    (e) Zeros    (f) Reshape

Figure 2: **Spectrum Error Evolution for the Generator.** We show one sample from the dataset in the first column and the mean and standard deviation of the reduced spectra for all 10 samples in the second column for reference. For the Toyset the dashed lines mark the mean of the Gaussian peaks at $0.1f_{nyq}$ and $0.8f_{nyq}$. Upsampling with bilinear or nearest neighbor interpolation biases the generator towards predicting little high-frequency content. Conversely, zero insertion and reshaping are prone to introducing checkerboard artifacts, indicated by the large errors at the highest frequency. The color corresponds to the relative error of the average predicted reduced spectrum wrt. the ground truth and is clipped at 1, i.e. when the relative error exceeds $100\%$.

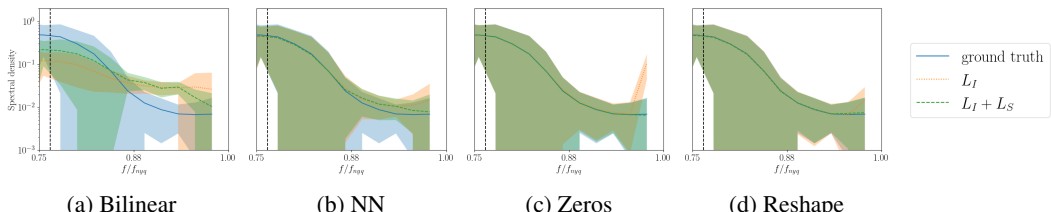

(a) Bilinear    (b) NN    (c) Zeros    (d) Reshape

Figure 3: **Reduced Spectrum for the Generator** on the Toyset. We plot the mean and standard deviation of the reduced spectrum above $0.75f_{nyq}$. The peak at the highest frequencies from upsampling with zero insertion or reshaping is removed by an additional loss on the spectrum.

objective, learnable filters can indeed compensate for high-frequency artifacts introduced by zero insertion and reshaping. Interestingly, bilinear upsampling does not benefit as much from the spectral loss. This suggests that a strong bias towards little high-frequency content can be more difficult to compensate for. As expected, the additional loss does not alter the evaluation in the spatial domain significantly and yields similar PSNR values in Table 1.

**Implications:** Overall, these results lead us to conclude that different upsampling operations bias the generator towards different spectral properties. Nearest neighbor and particularly bilinear upsampling introduce a bias towards fitting functions with little high-frequency content. In Section 5, we will see that this can also be beneficial when working with natural images as their spectral density follows an exponential decay. On the other hand, zero insertion and reshaping introduce a bias towards checkerboard artifacts. However, with a suitable loss function, the network filters can learn to compensate for these artifacts. Therefore, the upsampling in the generator alone cannot explain the spectral discrepancies. This suggests that the training signal provided by the discriminator might be suboptimal in the first place.

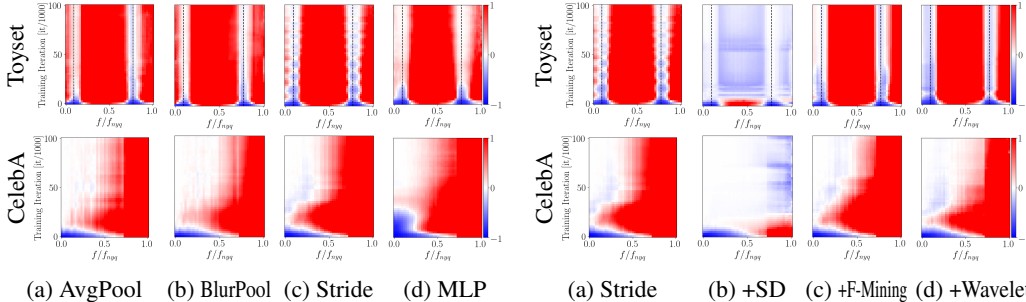

(a) AvgPool  (b) BlurPool  (c) Stride  (d) MLP       (a) Stride    (b) +SD  (c) +F-Mining (d) +Wavelet

Figure 4: **Spectrum Error Evolution for Discriminators with Different Downsampling Operations.** The downsampling operations do not significantly bias the discriminator towards any frequency range. Instead, it generally struggles with frequencies of low magnitude. The color corresponds to the relative error of the average predicted reduced spectrum wrt. the ground truth and is clipped at 1, i.e. when the relative error exceeds 100%.

Figure 5: **Spectrum Error Evolution for Discriminators on Different Input Domains.** The spectral discriminator greatly improves the spectral statistics on both datasets while hard example mining in the frequency domain (F-mining) and wavelets alter results only slightly. The color corresponds to the relative error of the average predicted reduced spectrum wrt. the ground truth and is clipped at 1, i.e. when the relative error exceeds 100%.

## 4 Can the discriminator provide a good training signal?

In this section, we investigate how good the training signal is that the discriminator can provide. Only a few existing works consider the discriminator as a cause for the spectral discrepancies. Chen et al. [5] attribute the high-frequency artifacts to information loss in the downsampling operations. In particular, they analyze how high frequencies affect the output of the discriminator at convergence. Instead, we consider how downsampling affects the training signal by assessing the input to the discriminator. Further, we analyze how the spectral statistics evolve during training to see if downsampling introduces a bias towards correcting low frequencies earlier than high frequencies. To reduce spectral discrepancies, Chen et al. [5] propose a regularizer based on the reduced spectrum to perform hard example mining in the frequency domain, cf. Eq. (3). Similarly, Jung et al. [15] define an additional discriminator on the reduced spectrum. While spectral discrepancies are not the main focus of their work, Gal et al. [10] also argue for "unfavorable loss functions" and propose to train both the generator and discriminator in wavelet space. In the second part of this section, we therefore aim to understand what makes (additional) inputs from other domains valuable and which of the proposed measures are most effective to correct the high-frequency artifacts.

**Experimental Setting:** Similar to the generator experiments in Section 3, we propose a conditional reconstruction task to assess the quality of the training signal independently of the generator architecture. More specifically, we train a class-conditional GAN with a single sample per class. To minimize the impact of the generator, we directly optimize the pixel values of the fake images. In practice, we pair 10 images with 10 labels as training data and optimize 10 learnable tensors conditioned on the labels. We optimize the learnable tensors and discriminator weights in an alternating fashion using the GAN two-player game setting. Similar to Section 3, we use the PGAN discriminator [16] with a reduced number of channels because our primary focus is on the spectral properties. As downsampling operations, we investigate strided convolution, average pooling, and blurring with subsequent average pooling [37]. We further train an MLP on the flattened input image as a baseline without any downsampling operations. When adding a spectrum discriminator, we weigh both discriminator losses equally as in [15]. Note, that training a GAN on such few images is a non-trivial task that requires careful tuning to train stably. We ensured that this is the case for our results, see supplementary for details.

**Datasets and Metrics:** We use the same datasets and metrics as in Section 3 and evaluate them on the reconstructions guided by the gradients from the discriminator. This allows us to assess the quality of the training signal.

**How does downsampling affect the training signal?** Fig. 4 shows the evolution of the average spectrum of the reconstructed images. On the Toyset, both the low- and high-frequency peaks are learned approximately at an equal pace. This suggests that the discriminator can indeed detect and correct the spectral statistics at high frequencies, regardless of the choice of downsampling operation. However,

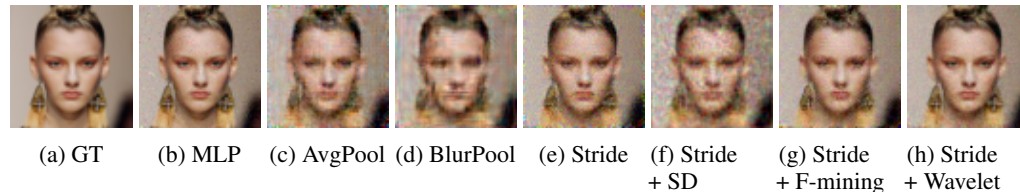

| (a) GT | (b) MLP | (c) AvgPool | (d) BlurPool | (e) Stride | (f) Stride + SD | (g) Stride + F-mining | (h) Stride + Wavelet |

Figure 6: **Reconstruction Guided by the Discriminator.** All discriminators with downsampling lead to significant deviations from the ground truth. While F-mining and wavelets slightly improve the reconstruction, the spectral discriminator decreases image fidelity.

all approaches struggle to correct frequencies of low magnitude. For natural images, the power spectrum decays exponentially which creates the impression that the discriminator struggles with high frequencies, while it actually struggles with the low magnitude of the high-frequency content. In image space, amongst the downsampling operations strided convolution achieves the best PSNR. However, it is much lower than the values for the generator in Table 1 which reflects the harder task due to instance- instead of pixel-level supervision. Fig. 6 shows the learned tensors at the last training iteration. While the reconstruction from the MLP is reasonably good, the downsampling operations introduce artifacts in the training signal provided by the discriminator.

**Is it enough to train in the spatial domain?** In Fig. 5 we compare the spectrum evolution for an additional spectral discriminator (SD) [15], hard example mining in the frequency domain (F-Mining) [5] and training in wavelet space (Wavelet) [10]. Since the spectral discriminator is directly applied to the reduced spectrum, it can guide the generator on the spectral statistics of the dataset and greatly reduces the error in Fig. 5b. Instead, F-mining and wavelets only slightly reduce the spectral discrepancies, e.g., on CelebA for frequencies between $0.5 - 0.75 f_{nyq}$. While F-mining increases the weight of samples with poor spectral realness, it does not make the discriminator more sensitive to slight changes in image space. This is in agreement with the findings in [5]. Wavelets separate the input wrt. the frequency but not wrt. the magnitude which could explain why low magnitude artifacts remain difficult to detect. In the spatial domain, however, the reconstructions with the spectral discriminator in Fig. 6e and 6f reveal a caveat: While the spectrum is better aligned, the reconstruction with the additional spectral discriminator qualitatively becomes worse. As the spectrum computation and the azimuthal integration discard information, images with the same reduced spectra can look very different. Consequently, penalizing the reduced spectrum might not be sufficient for improving the training signal alone. This also reflects in the PSNR in Table 2 which, except for the MLP, remains largely unaffected by the spectral discriminator. In the supplementary, we verify that replacing the reduced spectrum with the full Fourier transform indeed improves the image fidelity. However, as the spectral discriminator is an MLP, this does not trivially scale to real-world settings.

**Implications:** In contrast to existing hypotheses, our findings evidence that the discriminator generally struggles with frequencies that have a low magnitude but that high frequencies are not per se more difficult to detect. An additional discriminator on the reduced spectrum can greatly facilitate learning the spectral statistics of the data but might not improve the image fidelity. Even in our simple testbed, none of the convolutional architectures with downsampling is able to provide artifact-free supervision. This indicates that the discriminator might play a more important role in reducing high-frequency artifacts than currently anticipated in the field. While a simple spectral discriminator provides a good first step, we conclude that more work in this area is required to solve the problem. We believe that one key is to reduce the downsampling artifacts, e.g., by exploring alternative downsampling operations or by considering approaches that allow for pixel-level supervision of the discriminator as in [30] or [28].

## 5   Improving GANs

We now analyze the full GAN training, i.e., when generator and discriminator are two competing neural networks. First, we extend the discriminator analysis from Section 4 and verify if these results are also valid for full GAN training. Next, we investigate if the most effective discriminator is able to resolve high-frequency artifacts of the different upsampling strategies discussed in Section 3. Lastly, we analyze its effect on StyleGAN2 [19], a state-of-the-art GAN model, which shows the characteristic peak at the highest frequencies in the reduced spectrum.

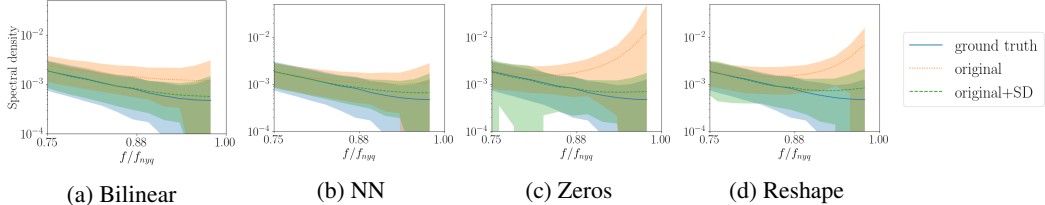

| (a) Bilinear | (b) NN | (c) Zeros | (d) Reshape |

Figure 7: **Reduced Spectrum for PGAN** on FFHQ64. We plot the mean and standard deviation of the reduced spectrum above $0.75 f_{nyq}$. The spectral discriminator prevents the high peak for zero insertion and reshaping but cannot fully correct the spectral discrepancies. Similar to Fig. 3, bilinear upsampling struggles to match the spectral statistics of the dataset.

| | Original | | Wavelet | | F-Mining | | SD | |
| --- | Acc | FID | Acc | FID | Acc | FID | Acc | FID |
| FFHQ64 | 72 | 31.0 | 73 | 39.8 | 73 | 33.7 | 62 | 31.0 |
| Cats128 | 88 | 106.1 | 74 | 122.5 | 77 | 119.3 | 62 | 114.6 |

Table 3: **Spectral Classification Accuracy and FID** for PGAN with discriminators on different input domains.

| | Bilinear | | NN | | Zeros | | Reshape | |
| --- | Acc | FID | Acc | FID | Acc | FID | Acc | FID |
| FFHQ64 | 57 | 36.3 | 62 | 31.0 | 66 | 34.9 | 67 | 35.2 |
| Cats128 | 62 | 138.2 | 62 | 114.6 | 77 | 128.8 | 70 | 129.7 |

Table 4: **Spectral Classification Accuracy and FID** for PGAN using different upsampling strategies with spectral discriminator.

**Experimental Setting:** We start by combining the architectures from Section 3 and Section 4 to obtain PGAN [16] with a reduced number of channels, which we train from scratch with R1-regularization and without progressive growing [19, 24] until the discriminator has seen 15M images. Next, we extend our analysis to StyleGAN2 on resolutions up to $1024^2$ pixels. The StyleGAN2 generator uses bilinear upsampling which, according to our previous findings, should not cause the elevated amount of high frequencies. Therefore, we finetune pre-trained models with the most effective discriminator following the training protocol from [17] until the discriminator has seen 2.5M images.

**Datasets:** We consider large-scale real-world datasets in this section. We train our version of PGAN on a downsampled version of FFHQ [18] at resolution $64^2$ pixels and a downsampled version of 200k images from LSUN Cats [36] at resolution $128^2$ pixels. We finetune StyleGAN2 on LSUN Cats ($256^2$ pixels), AFHQ Dog [6] ($512^2$ pixels) and FFHQ ($1024^2$ pixels). For AFHQ Dog, we use adaptive discriminator augmentation due to the small size of the dataset [17].

**Evaluation Metrics:** In the spatial domain, we report FID [12] on the full dataset and 50k generated images. To measure spectral discrepancies we deploy the spectral classifier described in Section 2. We follow the proposal in [4] and replace the KNN classifier with an SVM to obtain a stronger classifier. We use 1000 real and fake images each and use 90% and 10% for training and evaluation, respectively. Here, a classification accuracy of around 50% is ideal as this indicates a perfect generator because the classifier is forced to make a random guess.

**How do the isolated settings transfer to full GAN training?** We ablate PGAN with discriminators on different input domains in Table 3. In agreement with the findings from Section 4, the wavelet discriminator (Wavelet) and hard frequency mining (F-Mining) improve the spectral statistics only slightly. Hence, generated images can still be classified with high accuracy. The most effective method to learn the spectral statistics remains the additional spectral discriminator (SD) as indicated by the lower accuracy of the spectral classifier on all datasets. Consistent with our observation on the testbed, the image quality in the spatial domain remains largely unaffected.

| | Original | | +SD | |
| --- | Acc | FID | Acc | FID |
| Cats256 | 92 | 6.5 | 80 | 8.4 |
| AFHQ Dog | 92 | 7.4 | 66 | 12.1 |
| FFHQ | 99 | 2.8 | 94 | 3.1 |

Table 5: **Finetuning StyleGAN2** with an additional disciminator on the reduced spectrum.

Recalling the implication of Section 3, that the generator can learn to compensate for high-frequencies artifacts given a suited training objective, we now investigate whether the spectral discriminator satisfies such a requirement for different upsampling operations in the generator. Consistent with our observation on the generator testbed, Fig. 7 shows that the spectrum discriminator is also able to significantly reduce the peak at the highest frequency for both zero insertion and reshaping upsampling. However, the magnitude at the highest frequencies remains slightly elevated because the generator only receives supervision through the discriminator (real vs. fake) instead of full ground truth spectra considered in

the testbed. On the other hand, the bias towards little high-frequency content for bilinear and nearest neighbor upsampling aligns well with the spectral statistics of the datasets. This also reflects in Table 4 where images generated with zero insertion and reshaping are still detected with higher accuracy than images generated with bilinear and nearest neighbor upsampling. Considering both the spatial statistics and image fidelity, we observe that upsampling with nearest neighbor yields the best performance.

Lastly, we evaluate the effect of the spectral discriminator when applied to StyleGAN2 on datasets with higher resolution, namely LSUN Cats ($256^2$), AFHQ Dog ($512^2$) and FFHQ ($1024^2$). The results in Table 5 show mixed results: On AFHQ Dog the spectral discriminator results in a strong improvement on the spectrum but also significantly increases the FID. In contrast, on FFHQ the FID stays similar but the spectral discriminator also only slightly improves the spectral statistics and is not able to correct the peak at the highest frequency[1]. This suggests that in existing architectures there is a tradeoff between perceptual image quality and matching the spectral statistics of the data.

**Implications:** Our experiments show that both of our testbeds on the generator and the discriminator provide consistent observations which are aligned with those for full GAN training. Our findings suggest that the spectral discriminator can reduce the spectral discrepancies but the misalignment in the high frequencies is not fully addressed. We further observe that nearest neighbor upsampling together with an additional spectral discriminator is a good combination for natural images where high-frequency content is low. However, when applying existing measures to StyleGAN2, we observe that none of them can solve the spectral discrepancies completely. This suggests that the high-frequency artifacts in GANs are still an open problem that require further research. Our work takes an important step towards understanding the underlying mechanisms and sheds light on the effectiveness and limitations of existing approaches.

## 6 Other Related Work

**Deepfake Detection:** In the last years, great progress on unconditional Generative Adversarial Networks enabled photo-realistic image synthesis with deep neural networks [1,11,16,18,19,24,31]. With the rapid development in image synthesis, fake image detection becomes equally important [23,27,34]. Albeit very high photorealism, multiple works show that generated images can still be easily distinguished from real images based on their high-frequency content [7–9,14,22,34,38]. Consistent behavior across various architectures suggests a systematic problem in state-of-the-art GANs. Thus, it is important to understand if there is a frequency bias in convolutional generators or discriminators.

**Spectral Bias:** Fully connected ReLU networks are known to fit low-frequency modes of a target function faster than its high-frequency modes [2,3,13,26,35]. This behavior, referred to as *spectral bias*, has been shown to greatly impact the practical expressiveness of such models due to frequency dependent learning rates [2,3,13,26,35]. For CNNs, a spectral bias is often mentioned to explain the good generalization of largely overparameterized networks [32,33,35] but is theoretically less well understood. Inspired by Rahaman et al. [26] who study the spectral bias of fully connected ReLU networks, we analyze if a similar bias exists in convolutional generators and discriminators. Note that Rahaman et al. [26] consider one-dimensional regression tasks in which the spectrum is evaluated over the *output samples*. Instead, we calculate the spectrum over *all pixels* of one image. To highlight this difference, we use the term *frequency bias* in our work instead.

## 7 Conclusion

In this work, we take a thorough look at existing explanations for systematic artifacts in the spectral statistics of generated images and unify the efforts that have been done so far. While our experiments suggest that upsampling introduces a frequency bias in the generator this alone does not explain the spectral discrepancies. In agreement with [5,15], we find that advancing the discriminator brings us one step closer to learning the correct data distribution. However, none of the existing measures can faithfully recover the spectral statistics yet, such that generated images are still straightforward to classify solely based on their spectra. While state-of-the-art GANs benefit from highly engineered generator architectures, e.g., [16,18,19], our findings suggest that the design of the discriminator

---

[1]Jung et al. [15] conduct a similar experiment but evaluate the average over the logarithmic reduced spectra, i.e., $\frac{1}{N}\sum_{i=1}^{N}\log(\tilde{S}_i)$ while we consider $\frac{1}{N}\log\left(\sum_{i=1}^{N}\tilde{S}_i\right)$. We find that the peak at the highest frequencies is only significant in the latter computation and confirmed this difference with the authors.

plays an equally important role and deserves more attention in future work. We encourage future work to explore alternative downsampling operations and to consider approaches that enable pixel-level supervision for the discriminator. Interesting starting points could for example be [30] or [28]. Another interesting direction might be to explore discriminator augmentation [17, 39] wrt. high-frequency artifacts. We believe that our testbeds for both the generator and discriminator may serve as useful tools for future investigations and will make our code publicly available upon acceptance. Lastly, we remark that working on generative models always bears the risk of manipulation and the creation of misleading content. While the insights from our work could be used to make Deepfakes harder to detect, we believe that a better understanding of existing architectures and their limitations is key to advancing the field, such that the benefits of our work outweigh the societal risks.

## Acknowledgments

We acknowledge the financial support by the BMWi in the project KI Delta Learning (project number 19A19013O) and the support from the BMBF through the Tuebingen AI Center (FKZ: 01IS18039A). We thank the International Max Planck Research School for Intelligent Systems (IMPRS-IS) for supporting Katja Schwarz. This work was supported by an NVIDIA research gift.

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
