# OpenReview forum: "On the Frequency Bias of Generative Models"
_NeurIPS.cc/2021/Conference — NeurIPS 2021 Poster_

### Official Review · Reviewer_GSWk · 2021-07-13

**Rating:** 6
**Confidence:** 4

**Summary:**

Generative models, such as GANs, are known to not reproduce the spectral statistics. Various methods in the generator and discriminator have been proposed in papers. This paper systematically indexes and studies these tools and characterizes their effect on the spectral statistics of real vs generated images. The paper aims to use these insights to improve GAN training, with mixed results.

**Limitations And Societal Impact:**

Yes

**Main Review:**

This paper performs a systematic study of tools for both the generator and discriminator. The paper uses a "toyset", which consists of synthetically generated images with two dominant frequencies, as well as low-resolution faces (64x64). The paper first tests on small datasets (10 images). The paper focuses on the generation in Section 3 and the discriminator in Section 4. Section 5 attempts to draw upon these lessons to improve GANs. The paper evaluates on how well the generations reproduce the spectral properties, as well as image generation metrics like FID. The paper also presents on "detectability" from a classifier trained on spectral properties. The paper's presents a clear story and focus and systematically studies different factors.

I did have several concerns regarding the paper's presentation.
- I am confused regarding the colorbar in Figures 2, 4, 5. First, I recommend explaining the heatmap in the caption, rather than buried in the text. In L145, the text mentions the colorbar is clipped between -1 and +1. I understand how a frequency can be overrepresented by +100%, but how can a frequency can be underpresented by more than 100%? In the worst case, if a frequency is not represented, it should be at -100%? Furthermore, most of the plot is either clipped at solid blue or red. This seems quite surprising that almost none of the frequencies are even close.
- Figure 3 shows error bars, which adds to the value of the plot. However, the size of the error bars seem quite large. It is also quite difficult to tell the error bars apart, as they are heavily overlapping. Is it possible that standard deviation, rather than standard error, was plotted?

Minor
- Table 1. "39.81" has an extra significant figure
- L115 "investigatesnor"
- Equation 2: x used for H and y used for W is nonintuitive (to me)

While the paper systematically sudies different issues, the conclusions in this paper are not too impactful. As described in L170-175 and L229, many of the findings are in agreement with previous literature. This is good corroboration, but ideally this paper would have lead to new insights that could improve GAN training. In Section 5, the attempt at improving GANs are mixed. Lastly, while the paper focuses on quantitative results, there are not many qualitative results. The most impactful potential result would be clear, qualitative examples, where a more accurate spectral result provides more compelling generations as well.

In conclusion, I appreciate the systematic, focused study of this aspect of image generation. As summarized above, I have some concerns regarding the presentation; the paper does not show improvement in GAN training, and it's unclear if the paper's will lead to future improvements. I am currently borderline on this paper.

**Time Spent Reviewing:**

4

---

> ### Author Response · Authors · 2021-08-10
> **Response to Reviewer GSWk**
>
> Thank you for your constructive comments and valuable concerns. We hope we can clarify the presentation and scope of our work below:
>
> > I am confused regarding the colorbar in Figures 2, 4, 5. First, I recommend explaining the heatmap in the caption, rather than buried in the text. In L145, the text mentions the colorbar is clipped between -1 and +1. I understand how a frequency can be overrepresented by +100%, but how can a frequency can be underpresented by more than 100%? In the worst case, if a frequency is not represented, it should be at -100%?
>
> The largest possible negative error is indeed -1. We double-checked that this is the case for our evaluation (without clipping at -1) and will correct the formulation in  L.145 of the paper. Further, we will follow your suggestion and add the following explanation to the captions of Figures 2,4,5:
>
> *The color corresponds to the relative error of the average predicted reduced spectrum wrt.  the ground truth and is clipped at 1, i.e. when the relative error exceeds 100%.*
>
> > Furthermore, most of the plot is either clipped at solid blue or red. This seems quite surprising that almost none of the frequencies are even close.
>
> The clipping at solid red mostly occurs for the discriminator, i.e. Figure 4, 5, because it indeed struggles to closely match the distribution of frequencies with low magnitude. The generator with MSE-loss matches the spectral statistics more closely which results in little clipping at solid red and few solid blue values. One exception is the variant with bilinear upsampling on the Toyset for which the generator also struggles to learn the correct spectral distribution due to its stronger bias.
>
> > Figure 3 shows error bars, which adds to the value of the plot. However, the size of the error bars seem quite large. It is also quite difficult to tell the error bars apart, as they are heavily overlapping. Is it possible that standard deviation, rather than standard error, was plotted?
>
> We indeed plot the standard deviation over the 10 training images and clarify this in the caption of Figure 3. This is in agreement with other works in the field, see e.g. Dzanic et al. [1], Jung et al. [2].
>
> [1] T. Dzanic, K. Shah, and F. D. Witherden. Fourier spectrum discrepancies in deep network generated images. In Advances in Neural Information Processing Systems (NeurIPS), 2020.
>
> [2] S. Jung and M. Keuper. Spectral distribution aware image generation. In Proc. of the Conf. on Artificial Intelligence (AAAI), 2021.
>
> > While the paper systematically sudies different issues, the conclusions in this paper are not too impactful. As described in L170-175 and L229, many of the findings are in agreement with previous literature. This is good corroboration, but ideally this paper would have lead to new insights that could improve GAN training. In Section 5, the attempt at improving GANs are mixed. Lastly, while the paper focuses on quantitative results, there are not many qualitative results. The most impactful potential result would be clear, qualitative examples, where a more accurate spectral result provides more compelling generations as well.
>
> While our findings are in agreement with existing literature they also provide profound insights by considering the frequency bias of the network architectures. We demonstrate that the generator is able to learn the correct spectral distribution and that therefore the upsampling operations cannot explain the spectral discrepancies alone. This is an important insight because the majority of works in the field attribute the high-frequency artifacts to the upsampling operations [3,4,5,6,7]. Only a few works focus on the discriminator and, as evidenced by our experiments, none of them can fully resolve the high-frequency artifacts yet.
> We agree that finding a fix to improve GAN training is highly desirable and believe that our work takes an important step towards this goal by developing testbeds for both the generator and the discriminator and by providing a better understanding of the cause of the high-frequency artifacts.
>
> [3] K. Chandrasegaran, N. Tran, and N. Cheung. A closer look at fourier spectrum discrepancies for cnn-generated images detection. In Proc. IEEE Conf. on Computer Vision and Pattern Recognition (CVPR), 2021.
>
> [4] R. Durall, M. Keuper, and J. Keuper. Watch your up-convolution: CNN based generative deep neural networks are failing to reproduce spectral distributions. In Proc. IEEE Conf. on Computer Vision and Pattern Recognition (CVPR), 2020.
>
> [5] J. Frank, T. Eisenhofer, L. Schönherr, A. Fischer, D. Kolossa, and T. Holz. Leveraging frequency analysis for deep fake image recognition. In Proc. of the International Conf. on Machine learning (ICML), 2020.
>
> [6] A. Odena, V. Dumoulin, and C. Olah. Deconvolution and checkerboard artifacts. Distill, 2016.
>
> [7] X. Zhang, S. Karaman, and S. Chang. Detecting and simulating artifacts in GAN fake images. In IEEE International Workshop on Information Forensics and Security, 2019.

---

### Official Review · Reviewer_xGhA · 2021-07-13

**Rating:** 7
**Confidence:** 3

**Summary:**

The authors explore spectral properties of models in GAN training, isolating the generator and discriminator in turn, to address questions about inability for current frameworks to model certain data.


**Limitations And Societal Impact:**

Yes

**Main Review:**

-- The conclusion about the generator's ability to match spectral properties under the right setting is critical. I think the current assumption places much of the burden on high-frequency checkerboard artifacts on the generator's upsampling. To show the biases may exist, but that under some loss regimes the generator can learn to correct for them, is a very important observation.

-- The shift of focus both to the discriminator and to the magnitude of frequencies rather than if they are high or low is interesting, and motivates the adoption of alternative different discriminator architectures than otherwise would be explored.

-- A minor quibble: the authors claim that the MSE-based pixel-level supervision baseline of studying the generator are neutral and don't impart any biases of its own, e.g. with the claim 'otherwise, even with a perfect discriminator it would be impossible to generate a correct image'. While pixel-level supervision has the perfect minimum, it is possible that taking gradient descent steps toward it along a MSE-minimizing path will not reach that minimum while some other loss could. This is still a good baseline to use and I don't object to any of the conclusions drawn with it, making my disagreement very minor indeed.

-- One more question I would have liked to see explored is the impact on the distribution-level modeling of the data distribution. While obviously the point of this study is analyzing an individual point/image for its fidelity to a real point/image, I am curious about these statistics at a distribution level across the entire dataset. For example, if generator image i has particular artifacts, are those artifacts sensitive to what other points are in the training set (e.g. how diverse it is)? Or is that truly fundamental to the particular image i? To be clear, I understand this is a large question that need not be answered here, but it would have been interesting to see explored.

I think this work overall makes many important points that will positively impact the direction of future research. It is well-written and well-supported, and both raises and then answers interesting questions.


**Time Spent Reviewing:**

1

---

> ### Author Response · Authors · 2021-08-10
> **Response to Reviewer xGhA**
>
> Thank you for your helpful feedback and insightful questions. We hope we can clarify your questions below.
>
> > A minor quibble: the authors claim that the MSE-based pixel-level supervision baseline of studying the generator are neutral and don't impart any biases of its own, e.g. with the claim 'otherwise, even with a perfect discriminator it would be impossible to generate a correct image'. While pixel-level supervision has the perfect minimum, it is possible that taking gradient descent steps toward it along a MSE-minimizing path will not reach that minimum while some other loss could. This is still a good baseline to use and I don't object to any of the conclusions drawn with it, making my disagreement very minor indeed.
>
> We agree that an MSE-loss in combination with gradient descent does not necessarily lead to a perfect reconstruction. The goal of our experiments with pixel-level supervision is to study the bias of the generator architecture and to see if there is a setting in which the generator is able to compensate for spectral artifacts. Empirically, we find that an MSE-loss (on the image and optionally the reduced spectrum) is sufficient to demonstrate the generator’s ability to compensate for spectral artifacts.
>
> > One more question I would have liked to see explored is the impact on the distribution-level modeling of the data distribution. While obviously the point of this study is analyzing an individual point/image for its fidelity to a real point/image, I am curious about these statistics at a distribution level across the entire dataset. For example, if generator image i has particular artifacts, are those artifacts sensitive to what other points are in the training set (e.g. how diverse it is)? Or is that truly fundamental to the particular image i? To be clear, I understand this is a large question that need not be answered here, but it would have been interesting to see explored.
>
> We measure the impact on the data distribution indirectly by calculating FID and also consider the spectral distribution over the full dataset by plotting the mean and standard deviation of the reduced spectrum. Our findings are consistent across datasets which indicates that the artifacts are largely independent of the training set. Instead, they are rather specific to the building blocks used in the network architectures.
> We agree that it would be interesting to explore the occurrence of artifacts on the distribution-level in future work, e.g. to see if there are areas in latent space which are particularly prone to resulting in images with spectral artifacts.

---

### Official Review · Reviewer_Us7c · 2021-07-15

**Rating:** 7
**Confidence:** 4

**Summary:**

This paper provides insights on GANs via various experiments:
* Upsampling in G leads to different spectral biases.
  * Both bilinear and nearest neighbor upsampling bias the generator towards predicting little high-frequency content.
  * Zero insertion introduces checkerboard artifacts, but they can be compensated by a loss function sensitive to them.
* D struggles with frequencies of low magnitude, not with detecting high frequencies.
* Downsampling in D can impair training signals, but not if high-frequency has high magnitude.
* D benefits from additional input in frequency domains.


**Ethical Concerns:**

Fine.

**Limitations And Societal Impact:**

Limitation: n/a
Impact: Yes.

**Main Review:**

Originality
* (+) The authors investigate G and D on isolated environments.
  * (+) G is given reconstruction task with L2 loss on RGB and spectrum to penalize overall spectrum.
  * (+) Signals from D is evaluated by directly optimizing the pixel values.

Quality
* (+) The proposed experiments and implications are technically sound.
* (+) The paper provides ablation study from the isolated setting to the practical setting.

Clarity
* (+) Writing and organization are clear for understanding.

Significance
* (+) Many empirical evaluations are insightful for future research.
* (-) Most experiments are done with ProgressiveGAN and only the last part works with StyleGAN2.

Misc.
* L115 investigatesnor

**Time Spent Reviewing:**

1.5

---

> ### Author Response · Authors · 2021-08-10
> **Response to Reviewer Us7c**
>
> Thank you for reading and reviewing our work. We appreciate your feedback and that you find our work technically sound and insightful.
>
> > Most experiments are done with ProgressiveGAN and only the last part works with StyleGAN2.
>
> For our studies in Section 3 and 4 of the paper, we intentionally chose the architecture from ProgressiveGAN because of its simple architecture comprising only convolutions and upsampling operations. This allows us to directly analyze common building blocks of generative models, which are also used in state-of-the-art GAN methods like StyleGAN2. In Section 5 of the paper and Section 4.3 of the supplementary, we verify that the spectral artifacts indeed persist in StyleGAN2. Similarly, our findings on SNGAN in Section 4.2 of the supplementary suggest that our conclusions for the discriminator are consistent across architectures.

---

### Official Review · Reviewer_hXku · 2021-07-17

**Rating:** 7
**Confidence:** 4

**Summary:**

The paper studies the potential causes of  discrepancy in high frequency statistics between real and GAN-generated images. Specifically, it studies the upsampling choices in the generator, impact of downsampling in the discriminator and the corresponding frequency bias exhibited by such choices. The paper introduces clever experimental conditions to test various hypothesis related to the above study. Unsurprisingly, the authors claim that adding spectral matching loss or spectral discriminator helps match the spectral statistics. They don’t talk about the introduction of other artifacts in such conditions.  Authors conclude by saying that fully eliminating the discrepancy in the spectral statistics between real and generated distributions of images is still and open open problem. One of the key learnings from the paper is that the discriminators can provide good signals for learning high frequencies, but struggle to model frequencies with low magnitudes.


**Limitations And Societal Impact:**

The authors don't talk much about the limitations of the work.

**Main Review:**

Overall, I found the paper to be illuminating about how specific layer choices in generator and discriminator biases the learning of the distribution of generated images differently. The experiment design was quite clever and sufficient to substantiate various claims made in the paper. The paper is a good step towards understanding and solving the identified problems.

Some questions for the authors:
1. Authors choose 10 images for studying the choices made in discriminators and generators. What is the need of using such a small number of images? Will using more images forces the models to show their biases more clearly?
2. While correctly pointed out that there is a discrepancy in the generated and fake images, the goal of image synthesis is not just to match spectral statistics. What are the other side-effects introduced by spectral discriminator or spectral matching loss?


The paper is very easy to read and understand. The experiment design is very sound and convincing.

**Time Spent Reviewing:**

6

---

> ### Author Response · Authors · 2021-08-10
> **Response to Reviewer hXku**
>
> Thank you for the valuable comments and interesting questions.
>
> > What is the need of using such a small number of images? Will using more images force the models to show their biases more clearly?
>
> Following up on your question, we analyzed the biases of the generator and discriminator also for 100 and 1000 images. More specifically, for the generator, we investigate how more images impact the spectrum evolution for different upsampling types (bilinear, nearest neighbor, zeros, reshape) on the Toyset. The results and conclusions from this experiment are consistent with the results for 10 images in Figure 2 in the paper but the training time until convergence increases. For the discriminator, we investigate how more images impact the results for different downsampling strategies (AvgPool, BlurPool, Stride) with similar results and conclusions to Figure 4. We will add the aforementioned experiments on 100 and 1000 images to the supplementary.
>
> > While correctly pointed out that there is a discrepancy in the generated and fake images, the goal of image synthesis is not just to match spectral statistics. What are the other side-effects introduced by spectral discriminator or spectral matching loss?
>
> While a discriminator on the reduced spectrum can align the spectral statistics it might also reduce image fidelity, as can be seen in Figure 6 of the paper (Stride, Stride+SD) and in Figure 6 of the supplementary material. These side-effects can arise because the spectrum computation and the azimuthal integration discard information so that images with the same reduced spectra can look very different. In Section 3 of the supplementary, we verify that replacing the reduced spectrum with the full Fourier transform can remove these side-effects in combination with a fully-connected discriminator. As the fully-connected architecture does not scale to high image resolution, we further show that this approach cannot be trivially extended to remove the side-effects in real-world settings, i.e. when using a convolutional discriminator on the full Fourier transform.

---

### Decision · Program_Chairs · 2021-09-27

**Decision:**

Accept (Poster)

**Comment:**

The reviewers agreed that the paper was a strong systematic study of frequency bias of GANs. While the overall impact of the paper is somewhat limited, as previous work show similar results, it does provide sufficient new insight that it will be useful to the community.